# Non-invasive brain stimulation as treatment for motor impairment in people with Parkinson's disease: Protocol for an umbrella review

**Dale M. Harris** [1]*, **Steven J. O'Bryan** [1,2], **Christopher Latella** [3]

**1** Institute for Health and Sport (IHeS), Victoria University, Melbourne, Australia, **2** Institute for Physical Activity and Nutrition (IPAN), School of Exercise and Nutrition Sciences, Deakin University, Geelong, Australia, **3** Neurophysiology Research Laboratory, School of Medical and Health Sciences, Edith Cowan University, Joondalup, Australia

* dale.harris@vu.edu.au

## Abstract

**Data Availability Statement:** No datasets were generated or analysed during the current study. All relevant data from this study will be made available upon study completion.

### Background

Parkinson's disease (PD) is a progressive neurodegenerative disorder that predominantly affects movement and currently has no cure. Alongside medication, non-invasive brain stimulation (NIBS) may be used as an adjunct therapy to attenuate the motor symptoms experienced by people with PD. However, there is considerable heterogeneity in the evidence exploring the effects of NIBS for improving aspects of physical function in people with PD. Therefore, this protocol paper will outline the objectives, structure and procedure of a proposed umbrella review which will comprehensively summarise and map the current body evidence on the effectiveness of NIBS for improving physical function in people with PD.

### Methods and analysis

This study will adhere to the Joanna Briggs Institute (JBI) reviewer's manual and the PRISMA guidelines for conducting an umbrella review. The protocol is registered in PROS-PERO (CRD42022380544). The population, intervention, comparison, and outcomes (PICO) method will be used to guide the search strategies and inclusion/exclusion criteria. Systematic reviews, with or without meta-analyses, based on quantitative or mixed-methods studies, will be searched for, and then critically evaluated by two authors using the Assessment of Multiple Systematic Reviews 2 (AMSTAR2) tool. If the data allows, we will run a random effects pooled meta-analysis using standardized mean differences (SMDs), with heterogeneity and publication bias reported using the $I^2$ statistic. We will determine the level of evidence using the GRADE (Grading of Recommendations, Assessment, Development, and Evaluations) tool. Overlap in studies across reviews will be assessed using citation matrices and corrected covered areas (CCAs). Lastly, visual bubble plots will display the effects and strength of evidence from each review.

**Funding:** The authors received no specific funding for this work.

**Competing interests:** None.

## Discussion

This umbrella review will be the first to examine the collective evidence on the effectiveness of NIBS in improving physical outcomes for individuals with PD. It aims to provide an overall understanding of the relationship between NIBS and motor function changes, discuss underlying physiological mechanisms, and identify future therapeutic strategies.

## Trial registration

**PROSPERO registration number:** CRD42022380544.

## Introduction

Parkinson's disease (PD) is associated with debilitating postural instability and gait disturbance (PIGD) features [1], increasing the risk of falls and freezing of gait (FoG) episodes [2]. These PIGD motor features usually become less responsive to pharmacological treatment over time and are a major contributor to decreased mobility and quality of life (QoL), and increased mortality in people with PD. Moreover, the financial burden of drug and medical treatments escalates as the disease progresses, and falls and FoG episodes become more frequent [3]. As such, there is an urgent need for non-pharmacological interventions that target PIGD symptoms and mitigate the risk of falls and FoG in patients suffering from PD.

One non-pharmacological method which is generally effective in managing PIGD symptoms in PD is deep brain stimulation (DBS) [4,5]. However, DBS is invasive, highly selective based on disease severity and symptoms [6], and may occasionally exert detrimental effects including paradoxical worsening of pre-existing FoG or novel onset of FoG in some patients [7]. As an alternative approach, non-invasive brain stimulation (NIBS) has emerged as a promising clinical treatment option for people with PD in an attempt to restore neurological and physical function [8–11] and reduce FoG episodes [12]. Common NIBS techniques encompass transcranial direct current stimulation (tDCS), alternating current stimulation (tACS), random noise stimulation (tRNS), and repetitive transcranial magnetic stimulation (rTMS). Broadly, these techniques modulate neuronal activity in specific brain regions including the primary motor cortex, supplementary motor area, and dorsolateral prefrontal cortex, to alleviate motor symptoms and enhance motor function in PD. NIBS is relatively simple to administer, cost-effective, and has few side effects if treatment guidelines are followed [13]. However, the effectiveness of NIBS as a rehabilitative tool for people with PD is still uncertain due to considerable variability in outcomes and study design.

Over the past decade there has been a growing number of systematic reviews and meta-analyses focussed on the use of NIBS to improve physical outcomes in people with PD. However, significant heterogeneity in the findings likely due to the vast range of methods used to administer NIBS has limited any consensus on the efficacy of including the technique in the treatment of PD. Therefore, the primary aim of this umbrella mapping review is to summarise and map the overarching NIBS evidence on physical outcomes in people with PD to evaluate its feasibility as an effective treatment tool.

## Materials and methods

The protocol has been registered *a priori* in the international registry PROSPERO (ID: CRD42022380544). We have followed the Joanna Briggs Institute (JBI) reviewer's manual [14]

and adhered to the Preferred Reporting Items for Systematic Reviews and Meta-Analyses protocol (PRISMA-P) guidelines [15] (S1 Checklist). Additionally, we will employ the PICO (Population, Intervention, Comparison, and Outcomes) framework to guide our search strategy and inclusion/exclusion criteria.

*Population*: Eligible reviews will include trials with individuals diagnosed with idiopathic PD based on the Movement Disorder Society Clinical Diagnostic Criteria for PD (MDS-PD Criteria) [16], regardless of sex. Participants in these trials will range from stage 0 (no clinical signs present) to stage 4 (severe disability, but still able to walk and stand unassisted) according to the Hoehn and Yahr scale [17].

*Intervention*: The experimental intervention will apply either rTMS or tDCS, or other NIBS techniques (i.e., tACS, tRNS), as standalone therapies or in combination with exercise training, for more than a single session, over the dorsolateral prefrontal cortex, cerebellum, primary motor cortex, dorsal premotor cortex, or supplementary motor area. A single session of NIBS therapy will be defined as one uninterrupted application of the chosen stimulation technique of any length of time, either applied online (applied during a task or exercise) or offline (applied prior to performing a task or exercise).

*Comparison*: Within these reviews, the NIBS interventions are compared with sham stimulation, no treatment (i.e., regular care), or alternative treatment groups (i.e., exercise only, without sham stimulation). Sham NIBS will be defined as a simulated or placebo procedure that mimics the sensory experience of active NIBS interventions without inducing the intended neurophysiological effects. This often involves the application of a similar apparatus or procedure that creates sensory perceptions (e.g., tingling sensations or auditory clicks) but lacks the actual neuromodulatory impact of active stimulation.

*Outcome measures*: The effect of applying rTMS or tDCS, or other NIBS techniques, on physical outcomes in people with PD will be evaluated. Reviews will include section III of the UPDRS, or the timed-up-and-go test, as these are among the more common physical outcomes reported in PD research. Nevertheless, we expect a broad range of physical outcomes to be reported across studies. Therefore, where sufficient data are available, we will categorise physical outcome measurements into three groups: 1) motor function using the UPDRS section III, 2) functional movement (such as gait speed or dynamic balance), and 3) static balance or postural control measures.

*Study Design*: A comprehensive systematic review and meta-analysis of randomized controlled trials (RCTs) and controlled clinical trials (CCTs), that are written in English, and published in peer-reviewed journals within the past 10 years will be conducted. The systematic review should clearly outline a methodological approach for systematically searching, evaluating, and synthesizing data derived from the included studies.

## Inclusion criteria

When performing an umbrella review, using the same primary study across multiple systematic reviews can introduce bias by overemphasizing its significance [18]. This repetition inflates sample sizes and event counts, giving a false sense of precision. This can distort both narrative interpretations and statistical syntheses, affecting the accuracy and reliability of findings [18]. To mitigate this, we will select reviews from the past decade to minimise study overlap and avoid an overrepresentation of randomised controlled trials. We will include systematic reviews and meta-analyses of randomised and other controlled studies published from 1 December 2013 to 31 December 2023 comparing NIBS techniques (rTMS, tDCS, tACS, tNRS) with no NIBS intervention, exercise only, or sham stimulation in individuals with PD.

## Exclusion criteria

To maximise the number of studies which meet the inclusion criteria, no exclusions will be applied based on the presence of co-morbidities. However, during the screening process, if sufficient data is available, we will perform a sub-cohort analysis of the impact of NIBS on physical outcomes in participants with co-morbidities accompanying PD. We will exclude all systematic reviews (with or without meta-analyses) of randomised controlled trials or other experimental studies that were published before December 2013 (i.e., past 10 years of evidence), or were not written in English. For articles where the full-text was unavailable following an online search, attempts will be made to contact the corresponding authors via email to retrieve the full-text copy; if the full-text copy is not retrieved within a two-week period the article will be excluded from the analysis.

## Data sources and search strategy

Six separate databases will be searched: PubMed/MEDLINE, PEDro, Scopus, CINHL, EBSCO (i.e., PsychInfo, PsychTherapy and SPORTDiscus) and Cochrane Library, for systematic reviews (with or without meta-analyses) that were written in English up to December 2023. Various arrangements of the keywords "Parkinson's disease" AND "non-invasive brain stimulation or transcranial*" AND "physical function or strength or walking or gait" will be used within the title and/or abstract searches. Where a filter was absent, the key words in "systematic review", "meta-analysis" or "practice guidelines" will also be used within the title and/or abstract searches. A full list of search terms can be found in the S1 File. In addition, the reference lists of each of the included reviews will be manually checked to retrieve articles that were not covered by the database searches. A PRISMA flow chart will be developed to record the screening and selection of studies.

## Methodological quality

Methodological quality and bias risk will be recorded using the Joanna Briggs Institute critical appraisal checklist for Systematic Reviews and Research Syntheses [19]. For data extraction, three investigators (DMH, SJO and CL) will carry out the research autonomously, and subsequently cross-check the data to screen titles and abstracts using Covidence. The methodological quality of the included reviews will be assessed using the Assessment of Multiple Systematic Reviews 2 (AMSTAR2) instrument, which provides an empirical evaluation of their methodological rigor. The AMSTAR2 instrument employs a scoring system ranging from 0 to 16, with higher scores indicating higher quality [20]. It consists of 16 items related to various aspects of systematic reviews and meta-analyses, such as the comprehensiveness of the search strategy and the assessment of publication bias. Each item is scored dichotomously as either 0 or 1. Based on the AMSTAR2 scores, the quality of the reviews will be categorized as high quality (at least 80% of the items were satisfied), moderate quality (between 40% and 80% of the items were satisfied) or low quality (less than 40% of the items were satisfied) [20,21]. Once more, and to ensure consistency, all extracted review information and quality rankings will be compared among raters to establish inter-rater reliability.

We will also employ the Grading of Recommendations Assessment, Development, and Evaluation (GRADE) framework to assess the certainty of evidence quality across the outcomes of motor function (UPDRS-III), functional movement (timed-up-and-go and gait), symptom severity (freezing of gait questionnaire) and balance [22]. In summary, GRADE offers a systematic approach for evaluating the confidence in findings within meta-analyses, allowing for the determination of the strength of practical recommendations. Our assessment involved a modified GRADE scale, which considered factors such as bias risk, inconsistency,

indirectness, imprecision, and publication bias. Detailed information regarding the utilisation of this scale can be found elsewhere [22–24]. Following the GRADE evaluation, the quality of evidence is categorized as 1 = high strength of evidence; 2 = moderate strength of evidence; 3 = low strength of evidence; 4 = very low strength of evidence; or 5 = unable to determine the strength of evidence. Two authors will independently evaluate both the methodological quality and the quality of evidence. After completion, the scores will be compared between the authors, and any discrepancies will be thoroughly reviewed and adjusted.

## Meta-analysis

A random effect pooled meta-analysis will be performed whenever feasible, using the included effect sizes (standardized mean difference, SMD). By pooling the SMD's and applying 95% confidence intervals weighted for sample size, we will derive an aggregated effect size that encompasses all the reviews [25]. The estimated SMD's will be interpreted using the description presented by Hopkins et al. [26]; an SMD of >4.0 = an extremely large clinical effect; between 2.0–4.0 = a very large effect; 1.2–1.9 = a large effect; 0.6–1.1 = a moderate effect; 0.2–0.59 = a small effect; and 0.0–0.19 = a trivial effect. Heterogeneity among the reviews will be assessed using the $I^2$ statistic, which quantifies the degree of variability between study results ($< 25\%$ = negligible; 25–50% = low; 50–75% = moderate to high; $> 75\%$ = very high) [27]. Additionally, we will assess publication bias using the Egger test, which is a statistical method that evaluates the presence of bias in the reporting of study outcomes [28].

When the results from meta-analyses are reported as something other than SMD, for example, as mean difference (MD) or weighted mean difference (WMD), they will be re-expressed as SMD. To do this, we will enter in the primary study results to re-run the meta-analyses using JASP software (version 0.16.3) [29]. If necessary, confidence intervals (CI) and standard error (SE) will be converted to standard deviation (SD) using the formulas recommended by the Cochrane Handbook for Systematic Reviews of Interventions version 6.2 [30]:

$$SD = \sqrt{(N)}*(upper\ limit - lower\ limit)/3.92$$

OR

$$SD = \sqrt{(N)}*SE$$

Where N is the number of participants (sample size), and upper and lower bound limits refer to the 95% CI's, respectively.

## Analysis of degree of overlap in studies

Citation matrices will be generated to determine overlap in studies across reviews, while corrected covered areas (CCAs) will also be calculated (CCA = 0–5, slight; 6–10, moderate; 11–15, high; and >15, very high overlap) [31] to assess the impact of overlap in reviews on the findings of the umbrella mapping review.

## Evidence class

We will stratify the evidence using a classification method to allow for an objective, standardised classification of the level of evidence for each outcome. This classification will be based on strict criteria listed below, which has been previously recommended [32]. Convincing evidence (*class I*) will be determined when the number of cases exceeds 1000, p-value is less than 10^-6, $I^2$ is less than 50%, the 95% prediction interval excludes the null, there are no small-study effects, and no excess significance bias is present. Highly suggestive evidence (*class II*) will be

assigned when the number of cases exceeds 1000, the p-value is less than $10^{-6}$, the largest study shows a statistically significant effect, and *class I* criteria are not met. Suggestive evidence (*class III*) will be designated when the number of cases exceeds 1000, the p-value is less than $10^{-3}$, and *class I–II* criteria are not met. Weak evidence (*class IV)* will be assigned when the p-value is less than 0.05 and *class I–III* criteria are not met. Evidence will be considered non-significant when the p-value exceeds 0.05.

### Evidence map

A visual bubble plot of the scientific evidence of each included review will be designed to represent the effects and strength of each systematic review across four dimensions. Firstly, the number of participants within each review will be depicted proportionally by the size of the bubbles. Secondly, the NIBS mode (rTMS, tDCS, tACA, tNRS) will be indicated by the colour of each bubble. Thirdly, on the x-axis, systematic reviews will be classified based on the effects found: "unclear" if there was insufficient evidence, "no differences" if no differences were observed, and "mixed results" if contradictory outcomes were reported. Lastly, on the y-axis, reviews will be ranked according to their methodological rigor using the AMSTAR2 assessment criteria.

## Ethics and dissemination

Ethics approval is not required as data will be searched for and gathered based on the published systematic reviews and meta-analyses. We plan to publish the results of this umbrella review in a peer-reviewed journal. We will also present these results at a neurology or neurostimulation conference. All the relevant additional data will be uploaded to the online open access databases.

## Discussion

To the best of our knowledge, this marks the first umbrella review to be conducted focused on examining the collective evidence on the effectiveness of NIBS in improving physical outcomes among individuals with PD. From a clinical standpoint, gaining an overall understanding of the relationship between NIBS and changes in motor function is highly relevant for people with PD, their caregiver, and clinicians to help make informed treatment choices for managing PIGD or FoG symptoms alongside traditional PD medications.

This umbrella review will also aim to provide discussion around the underlying physiological mechanisms through which NIBS may contribute to the enhancement of motor function in people with PD. In addition, the review will endeavour to identify and highlight potential avenues for future therapeutic strategies that encompass NIBS as part of wider therapy. Importantly, the review will provide recommendations and guidance on the application of NIBS as part of physical rehabilitation treatment modality for individuals with PD. Indeed, NIBS is becoming more popular in clinical settings as part of treatment of motor and non-motor system problems [11]. Hence, it is possible that existing infrastructure, equipment, and skills are already in existence across some patient care settings (e.g., hospitals and rehabilitation clinics) which would help facilitate practitioner upskilling and therapeutic adoption.

Our umbrella review will have several strengths. Firstly, we are committed to rigorously adhering to the PRISMA guidelines throughout the review process. The methodological quality of the eligible systematic reviews and meta-analyses included in this umbrella review will be assessed using the AMSTAR2 tool, ensuring the robustness of the included reviews, and enhancing the reliability of the findings. Secondly, we aim to explore the potential biases and heterogeneity in the existing studies, which should help to strengthen the methodological

approaches taken in future NIBS research. We will also provide objective and standardised classifications of the level of evidence for each physical outcome reported, which will be helpful for clinicians in evaluating the strength of evidence presented and reduce the current heterogeneity in outcome measures. Lastly, our approach encompasses multiple outcome measures, with a focus on assessing the effects of NIBS across various dimensions of physical function in individuals with PD.

## Supporting information

**S1 Checklist. PRISMA-P (Preferred Reporting Items for Systematic review and Meta-Analysis Protocols) 2015 checklist: Recommended items to address in a systematic review protocol\*.**
(DOCX)

**S1 File.**
(DOCX)

## Author Contributions

**Conceptualization:** Dale M. Harris.

**Methodology:** Dale M. Harris.

**Writing – original draft:** Dale M. Harris.

**Writing – review & editing:** Dale M. Harris, Steven J. O'Bryan, Christopher Latella.

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
