## [Decision Letter · Decision Letter 0]

17 May 2024

Non-invasive Brain Stimulation as Treatment for Motor Impairment in People with Parkinson's Disease: Protocol for an Umbrella Review

PONE-D-24-08118

Dear Dr. Harris,

We’re pleased to inform you that your manuscript has been judged scientifically suitable for publication and will be formally accepted for publication once it meets all outstanding technical requirements.

Kind regards,

Vishal Jain

Academic Editor

PLOS ONE

Journal Requirements:

Additional Editor Comments (optional):

Reviewers' comments:

Reviewer's Responses to Questions

**Comments to the Author**

1. Does the manuscript provide a valid rationale for the proposed study, with clearly identified and justified research questions?

Reviewer #1: Yes

Reviewer #2: Yes

2. Is the protocol technically sound and planned in a manner that will lead to a meaningful outcome and allow testing the stated hypotheses?

Reviewer #1: Yes

Reviewer #2: Yes

3. Is the methodology feasible and described in sufficient detail to allow the work to be replicable?

Reviewer #1: Yes

Reviewer #2: Yes

4. Have the authors described where all data underlying the findings will be made available when the study is complete?

Reviewer #1: Yes

Reviewer #2: Yes

5. Is the manuscript presented in an intelligible fashion and written in standard English?

Reviewer #1: Yes

Reviewer #2: Yes

6. Review Comments to the Author

You may also provide optional suggestions and comments to authors that they might find helpful in planning their study.

Reviewer #1: This protocol is for an umbrella review which will be of high significance as it deals with non invasive brain stimulation as treatment for motor impairment in people with parkinson's disease. The authors have described the protocol in detail and it follows the PlosOne guidelines. The umbrella review proposed in this protocol will adhere to the JBI and PRISMA guidelines. The authors have done a good job in detailing the material and methods along with inclusion and exclusion criterion. They also talk about the statistical approached that they will take for the umbrella review. The authors also talk about the strength of the umbrella review.

The only aspect that the authors do not talk about is the shortcomings of the protocol. Every protocol has strengths and shortcomings and the authors should talk about the shortcomings of this protocol.

Reviewer #2: In the inclusive criteria, please also include studies so that it will cover a wide range of population across different regions of the world. This will help in making conclusions regarding the effectiveness of the method on different populations of people.

7. PLOS authors have the option to publish the peer review history of their article (what does this mean?). If published, this will include your full peer review and any attached files.

Reviewer #1: No

Reviewer #2: No

---

## [Editor Report · Acceptance letter]

24 May 2024

PONE-D-24-08118 

PLOS ONE

Dear Dr. Harris, 

I'm pleased to inform you that your manuscript has been deemed suitable for publication in PLOS ONE. Congratulations! Your manuscript is now being handed over to our production team.

Kind regards, 

on behalf of

Dr. Vishal Jain 

Academic Editor

PLOS ONE